# Genomic Profiling and Clinical Outcomes of Targeted Therapies in Adult Patients with Soft Tissue Sarcomas

**DOI:** 10.3390/cells12222632

**Published:** 2023-11-15

**Authors:** Stefania Kokkali, Eleni Georgaki, Georgios Mandrakis, Claudia Valverde, Stamatios Theocharis

**Affiliations:** 1First Department of Pathology, Medical School, National and Kapodistrian University of Athens, 75 Mikras Asias Street, 11527 Athens, Greece; giormandr@biol.uoa.gr; 2Oncology Unit, 2nd Department of Medicine, Medical School, Hippocratio General Hospital of Athens, National and Kapodistrian University of Athens, V. Sofias 114, 11527 Athens, Greece; elenigeorgaki@hippocratio.gr; 3Medical Oncology Department, Vall d’Hebron University Hospital, Pg. Vall d’Hebron 119-12, 08035 Barcelona, Spain; cvalverde@vhio.net

**Keywords:** next-generation sequencing, DNA sequencing, RNA sequencing, comprehensive genomic profiling, soft tissue sarcoma, genetic aberration, actionable alterations, targeted therapy, precision medicine

## Abstract

Genomic profiling has improved our understanding of the pathogenesis of different cancers and led to the development of several targeted therapies, especially in epithelial tumors. In this review, we focus on the clinical utility of next-generation sequencing (NGS) to inform therapeutics in soft tissue sarcoma (STS). The role of NGS is still controversial in patients with sarcoma, given the low mutational burden and the lack of recurrent targetable alterations in most of the sarcoma histotypes. The clinical impact of genomic profiling in STS has not been investigated prospectively. A limited number of retrospective, mainly single-institution, studies have addressed this issue using various NGS technologies and platforms and a variety of criteria to define a genomic alteration as actionable. Despite the detailed reports on the different gene mutations, fusions, or amplifications that were detected, data on the use and efficacy of targeted treatment are very scarce at present. With the exception of gastrointestinal stromal tumors (GISTs), these targeted therapies are administered either through off-label prescription of an approved drug or enrollment in a matched clinical trial. Based mainly on anecdotal reports, the outcome of targeted therapies in the different STS histotypes is discussed. Prospective studies are warranted to assess whether genomic profiling improves the management of STS patients.

## 1. Introduction

Soft tissue sarcomas (STS) constitute a group of heterogeneous tumors of mesenchymal origin that are classified into more than 100 histological types [1]. Patients with advanced or metastatic disease have a poor prognosis, with a median overall survival (OS) of approximately 1.5 years from the start of therapy [2]. Therefore, there is an unmet need for the development of new therapies. Tumor-agnostic precision medicine has been developed over the last decade, employing comprehensive genomic profiling. High-throughput, large-scale sequencing capacity of next-generation sequencing (NGS) technologies holds the promise of identifying genomic alterations that are pharmacologically tractable.

In STS, the translation of genomic discoveries to the clinic is still under investigation. Although molecular methods have been traditionally used in STS diagnostics [3,4], their potential application in theranostics is still debatable [5]. Notwithstanding the relatively high occurrence of clinically actionable biomarkers in different STS histotypes identified by recent NGS studies, the clinical benefit of these findings seems low [6,7,8]. The detection of molecular alterations with unknown effects on the specific sarcoma histotype is a major limitation. In this review, we summarize published reports on genetic alterations with therapeutic relevance detected by NGS in STS patient specimens, focusing on the efficacy of targeted therapies that have been administrated, either through off-label prescription of an approved drug or enrollment in a matched clinical trial.

## 2. Methods

Studies were identified by searching the PubMed electronic database for the following terms: (“comprehensive genomic profiling” OR “genomic profiling” OR “next-generation sequencing”) AND (“sarcoma” OR “soft tissue sarcoma” OR “MPNST”). Malignant peripheral nerve sheath tumor (MPNST) is considered a relatively common STS histotype and, therefore, was included in the search terms. Results were restricted to adult human studies and articles written in the English language. No year restriction was applied. All retrieved studies were reviewed and accepted for further processing if contained information on NGS-based therapy administration and efficacy. All reports were analyzed for genomic results informing choices regarding targeted therapy. Due to the scarcity of data, case reports were not excluded. Studies reporting results on all STS histotypes were included, except for gastrointestinal stromal tumors (GISTs), given the established role for molecular testing in this entity and the approved KIT- and PDGFRα-targeting drugs. Studies analyzing the clinical utility of genomic profiling in cancer patients in general were also included if specific information on sarcoma patients was identified.

## 3. Studies Reporting Sequencing Results Leading to Targeted Therapy Implementation across STS Histotypes Containing Efficacy Data

We identified 18 studies reporting the results of comprehensive genomic profiling along with the clinical impact of implemented targeted therapy in 61 STS patients (Table 1). These reports include mainly retrospective studies on NGS in sarcoma patients from comprehensive cancer centers in the USA (n = 6) [7,8,9,10,11,12] and also from China [13] and Portugal [14]. In addition, five prospective studies provide data on genomic alterations and matched treatments: one from the USA [15] and four from Europe [16,17,18,19]. Furthermore, a limited number of case reports illustrating clinical benefits from targeted therapies in STS patients were also identified [20,21,22,23]. The majority of studies analyzed only sarcoma patients, whereas two studies analyzed different tumor types [17,18] and one study included exclusively sarcomas of hematopoietic origins [11]. Finally, another two prospective studies screened cancer patients for a specific genomic aberration, *IDH* mutations [24], and *MET*ex14 alterations [25].

## 4. Occurrence of Clinically Actionable Genomic Abnormalities in Patients with STS 

Integrative genomic testing detects frequently genetic alterations in STS specimens. Base substitutions, short indels, and copy number variations (CNVs) are found in different genes, with *TP53*, *RB1*, and *CDKN2A/B* being among the most common ones [6,7,10,12,26,27,28]. Furthermore, gene fusions leading mainly to the activation of transcription factors constitute a common finding. The frequency of genomic abnormalities depends on the gene panel and the technique that has been employed. Analysis of genomic data from 584 adult STS patients included in the GENIE database revealed 2697 alterations in 451 genes [6]. In a series from the USA consisting of 133 sarcomas, Cote et al. reported a median number of 2 gene alterations per tumor, tested by targeted NGS (both DNA and RNA sequencing), including more than 400 cancer-related genes [27]. Comprehensive genomic profiling in 102 sarcoma patients from MD Anderson identified at least 1 genomic alteration in 93% of the samples, with a median number of 6 alterations per patient [10]. Similar results were reported for NGS testing (targeting 450 cancer-associated genes) in sarcoma patients from a single institution in China [28]. In another retrospective study from Florida, 96.7% of non-GIST sarcoma patients were found to harbor at least one alteration [7]. Sarcomas with complex genomics harbor more alterations compared to translocation-relative sarcomas, as illustrated by the example of leiomyosarcoma.

Subtype-specific genomic aberrations have been identified in several studies, including diverse STS histotypes [26], while studies investigating exclusively a specific histotype shed more light on the genomic landscape of these sarcomas [11,15,29,30,31,32,33]. Approximately 30–40% of sarcomas are characterized by a well-defined recurrent genetic alteration that contributes to its pathogenesis [34,35,36,37]. These alterations were detected using traditional molecular methods, such as fluorescence in situ hybridization (FISH) and polymerase chain reaction (PCR), and are valuable diagnostic tools, whereas most of them are not targetable. They include chromosomal translocations involving transcription factors, complex chromosomal aberrations, over-expression of receptor kinase ligands, inactivation of regulatory proteins, gene mutations, and gene amplifications, as exemplified by *CDK4/MDM2* amplification in liposarcoma. Modern sequencing techniques, such as NGS, revealed a large variety of additional histotype-specific alterations, occurring only in a proportion of patients with this sarcoma histotype. According to these data, mutations in *PIK3CA* have been reported in approximately 1/5 patients with myxoid/round cell liposarcoma [26,38]; alterations in *NF1* are found in a subset of patients with MPNST, myxofibrosarcoma, and pleomorphic liposarcoma [26,30], and *TP53* mutations are frequent in dedifferentiated liposarcoma (DDLPS), leiomyosarcoma and myxofibrosarcoma [12,28], and alterations in homologous recombination deficiency (HRD) genes in leiomyosarcoma [12]. There are plenty of other observations of STS subtypes characterized by specific genetic aberrations.

Making sense of the results generated by comprehensive genomic profiling and identifying therapeutic vulnerabilities is a difficult task. A significant proportion of the above genomic alterations were considered potentially actionable across the different studies based on corollary evidence obtained in other tumor entities or preclinical data. The question of clinical actionability is discussed within a molecular tumor board in most institutions. In the largest retrospective series from the USA, in which 7494 STS specimens were sequenced, 31.7% exhibited potentially actionable alterations [12]. A higher proportion of 41%, 47%, 49%, and 45% was reported from the GENIE database [6] and three single institutions in Chicago [8], Florida [7], and MD Anderson [10], respectively, whereas Cote et al. reported an even higher proportion of 88% [27]. Although all of these cohorts included more than 100 STS, the number of patients per specific histotype was small. However, some STS subtypes were identified to harbor a higher number of targetable alterations, such as leiomyosarcoma [17] and liposarcomas [10]. In a large genomic analysis in Germany, synovial sarcoma patients had the highest clinical benefit among STS, followed by liposarcoma and leiomyosarcoma patients [17]. Finally, in a smaller retrospective series of 65 STS in Japan, potentially actionable genomic alterations were found in 26 tumors (40%) [13]. It should be noted that most of these studies include a low proportion of GIST patients as well.

Genetic aberrations involving a single gene lead to loss or gain of gene function. Furthermore, composite biomarkers are also interrogated through NGS, such as tumor mutational burden (TMB). Genomic aberrations detected in STS can be assigned to the following cellular pathways and processes: (i) **tyrosine kinase** (TK) activation or inactivation, including *FGFR* amplification or fusion, *VEGF* amplification, *RET* amplification, *BRAF* mutation, *MET* amplification, *FGF* amplification, and *FRS2* amplification; (ii) **PI3K–AKT–mTOR** (PAM) **pathway**, including *TSC2* deletion or base substitution, *VHL* deletion, *NF2* deletion, *TSC1* deletion or base substitution or amplification, and *PTEN* deletion; (iii) **RAF–MEK–ERK** (RME) **pathway**, such as *KRAS* amplification or base substitution, *NRA* base substitution, *NF1*-inactivating mutation or deletion, *NF2* deletion, and *MAP2K2* amplification; (iv) **cell cycle**, such as *MDM2* amplification, *CDK4* amplification, *CDKN2A* deletion or base substitution, *CDKN2B* deletion, *MYC* amplification, *CDK6* amplification, *CCNE1* amplification, and *ARID1A* base substitution; (v) **DNA damage repair** (DDR), such as *BRCA1* and *BRCA2* deletions, *PTEN* deletion, *BAP1* deletion, *FANCE* fusion, *ATM* base substitution, and mismatch repair (MMR) deficiency (defined as loss or inactivation of *MLH1*, *MSH2*, *MSH3*, *MSH6*, *PMS2*); and (vi) **immune evasion**, including *TCTLA4* amplification, *PDCD1* amplification, and TMB [6,7,8,10,12,16,17,19,27,28,30]. As mentioned above, there are some associations of genomic alterations with specific STS subgroups. Alterations in cell cycle genes are more common in liposarcomas [17], MPNST harbor alterations in the RME pathway [30], and angiosarcoma in angiogenesis-related genes (TK family) [31]. TMB has been assessed across STS and has been found high in some histotypes, such as rhabdomyosarcomas [16], UPS [12], MPNST [12], and leiomyosarcomas [17]. Anecdotal cases of high TMB have also been reported in patients with angiosarcoma, fibrosarcoma, and unclassified sarcoma [15,28]. Homologous recombination deficiency (HRD), defined by alterations with loss of function in DDR genes, is found in a small subset of STS, mainly leiomyosarcomas. 

## 5. Only a Small Proportion of Patients Receive Matched Therapy

Notwithstanding the high frequency of potentially actionable genomic alterations observed across STS studies, only a minority of patients finally receive drugs targeting an alteration detected by genomic profiling. In a retrospective cohort (N = 118) from the Memorial Sloan Kettering Cancer Center (MSKCC), 29% of the patients either were enrolled in a relevant clinical trial or received off-label molecularly matched therapy [12]. In another cohort of sarcoma patients referred to the phase I clinical trials program at MD Anderson (N = 102), only 16% received a targeted therapy [10]. In another cohort from a single institution in the USA (N = 136), 8.8% of STS patients received a genomically driven therapy [8]. In another single-institution series in the USA (N = 114), NGS-based therapy was administered to 7% of advanced sarcoma patients [7]. In a smaller cohort of thirty-four sarcoma patients from a single institution in the USA, change in medical treatment was reported only in four patients (11.8%), despite the availability of matched clinical trials in 73% of patients [39]. Genomic analysis of 158 sarcoma patients, using a 69-gene panel, in a French center led to treatment implementation in 8.2% of patients [19], whereas in a subsequent molecular screening program in the same center (N = 39 sarcoma patients), this proportion is 17.8% [18]. In a small prospective study including 58 STS patients in another French center, 12% were enrolled in early phase clinical trials of matched drugs [6]. In a smaller retrospective study in Portugal (N = 30), 10% of sarcoma patients were treated with a genome-targeted drug [14]. In the prospective European EORTC proof-of-concept study of 71 adolescent and young adults with sarcoma, 2.8% of all patients received targeted therapy [16]. Finally, among 65 STS patients in China, 9.2% received matched therapy in a clinical trial [13].

Taken together, these observations highlight that the clinical translation of molecular profiling in sarcomas is still limited (Figure 1). A significant proportion of patients with a treatment recommendation do not receive targeted therapy for different reasons. A major limitation is the lack of access to or reimbursement of the recommended drug. Most of these drugs are either approved in another tumor type or investigational and can be delivered to patients through enrollment in relevant clinical trials (usually basket), compassionate use programs, or off-label use. In addition, patients usually undergo genomic profiling late during the disease course, with a number of them dying before the results of the molecular test or when their general condition has worsened. In some cases, the treating clinician decided to administer standard-of-care therapies instead of following the recommendation made by the molecular tumor board. Furthermore, some patients do not progress during the observation period of the studies; therefore, it is unknown whether the therapeutic selection is influenced by the molecular test in the future.

## 6. Efficacy Results of Targeted Therapies in Soft Tissue Sarcomas Patients 

Most studies on comprehensive genomic profiling in STS have provided a large amount of data on the occurrence of genetic alterations in STS, coupled with targeted therapy recommendations. The efficacy of these molecularly informed therapies is largely unknown to date. They have not been evaluated in prospective randomized trials, only in signal-seeking trials, with very scarce data on the outcome of patients who were treated with them in the advanced setting. Table 1 summarizes the outcomes of STS patients who were treated with therapies targeting genomic alterations identified by NGS. In total, sixty-nine patients were identified, of which fifteen were enrolled in seven prospective genomic profiling studies [15,16,17,18,19,24,25], and fifty are reported in the different retrospective series of NGS in STS [6,7,9,10,11,12,13,14]. Some studies also included transcriptomic analysis; treatment recommendations based on mRNA expression are excluded, as only therapies targeting genomic abnormalities are described in this review. In addition, anecdotal benefits of targeted therapy are described in four case reports [20,21,22,23]. Different STS histotypes are included, with liposarcoma being the most common (N = 14; five dedifferentiated liposarcoma (DDLPS), four well-differentiated liposarcoma (WDLPS), one myxoid liposarcoma, and four without a precise subtype), followed by leiomyosarcoma (N = 12), UPS (N = 7), and angiosarcoma (N = 6). There are also ultra-rare histotypes, such as epithelioid hemangioendothelioma (EHE), PEComa, clear cell sarcoma, histiocytic sarcoma and inflammatory myofibroblastic tumors (IMTs).

FoundationOne^®^ was the most common sequencing platform used in the studies that we analyzed [7,8,9,10,12,14,20,22,24,25]. In three studies, whole-exome sequencing was performed [15,16,17], whereas in two studies, two different commercial NGS platforms were used [8,13]. Additionally, NGS data were generated through in-house targeted sequencing platforms in three reports [6,11,19]. Oncomine Focus Assay^®^, including 52 genes, was used in a case report [23]. Finally, FoundationOne^®^ was compared to a smaller in-house platform in the French ProfiLER 02 trial [18]. The majority of STS patients received molecularly matched therapy in a drug-matched clinical trial [6,7,10,12,13,17,19,20,22,24]. Off-label targeted therapies were also administered in several patients [7,9,12,15,16,17,19,25], and a limited number of patients received targeted therapy on a compassionate use basis [17,21]. Access to matched therapy is not elucidated in four studies [8,11,14,18].

Palbociclib, a CDK4/6 inhibitor, was used in 13 STS patients harboring *CDKN2A* (N = 8), *CDK4* (N = 3), or *MYC* (N = 2) alterations [7,8,9,13]. In most cases, it was administered as a monotherapy, whereas in one patient, it was combined with the anti-estrogen fulvestrant [7]. The majority of patients treated with this agent originated from a single-institution series in the USA [8]. In twelve cases therapies targeting angiogenesis were implemented, as these tumors harbored alterations in angiogenesis-related genes [6,7,8,13,14,17,19]. Pazopanib was the most frequently used drug in this setting (N = 7), followed by FGFR inhibitors in three patients and imatinib in the remaining two patients. *BRAF* mutations were targeted in seven patients, with either a BRAF inhibitor in four patients [6,7,10,13] or a combination with an MEK inhibitor in three patients [11,12,20]. Furthermore, a patient with a *BRAF* fusion initiated combinational treatment with sorafenib/bevacizumab/temsirolimus [10]. Five patients with WDLPS/DDLPS, from another single institution in the USA, received MDM2 inhibitors, based on the well-known *MDM2* amplification of these tumors [10]. In another four cases, agents targeting *ROS1* aberrations were implemented, including ceritinib (N = 1), crizotinib/pazopanib combination (N = 2), and an investigational ALK/ROS1/NTRK inhibitor (N = 1) [7,10]. The NTRK inhibitor larotrectinib was used in two cases [12,23]. Lastly, seven patients were treated with immune checkpoint inhibitors (ICIs) driven by an intermediate (N = 3) [7,12,21] or high TMB (N = 4) [12,15,18]. Two of them received double inhibition with anti-PD1 + anti-CTLA4 monoclonal antibodies. 

Assessing for efficacy of the different targeted therapies implemented across STS patients is challenging, as they were given in different lines of treatment, usually in heavily pretreated patients, either in retrospective studies or early phase trials with varying follow-up periods. These factors, along with the predictive value of the selected biomarker, influence the efficacy of the drugs. Treatment outcomes are illustrated in Table 1, providing all the available information. Progression-free survival is provided only in some cases, whereas the best response is reported for the majority of patients. Data on outcomes are quoted for a qualitative estimate of the clinical benefit of NGS-based therapies in STS patients, instead of a statistical analysis of efficacy parameters. 

Five patients exhibited complete response (CR) or near CR as the best response to targeted therapies; double immunotherapy (N = 2) or ICI monotherapy (N = 1) for high or intermediate TMB [12,18], an MDM2 inhibitor in one patient with WDLPS [10], and larotrectinib, an NTRK inhibitor, in another patient with an inflammatory myofibroblastic tumor (IMT) harboring *ETV6*-*NTRK3* fusion [12]. Fourteen additional patients experienced a partial response to molecularly driven therapies. BRAF inhibitors, either as a monotherapy (N = 2) or in combination with an MEK inhibitor (N = 2), were the most common drugs leading to PR, irrespective of histology [6,10,11,12]. Different other drugs led to PR, including TKIs (crizotinib for *MET*ex14 alteration and *ALK* fusion, imatinib for *PDGFRa* mutation, pazopanib for *FLTA4* amplification, larotrectinib for *NTRK* gene fusion) [14,19,22,23,25], mTOR inhibitors for *IGF1R* and *PIK3CA* alterations [6,14], an MDM2 inhibitor for *MDM2* amplification [10], and tazemetostat, an epigenetic regulator, for *SMARCB1* deletion [12,40].

Progression-free survival (PFS) varies between 1.4 and 44.3 months. The longest PFS was noted with ICI in two angiosarcoma patients [15], with a PFS of 32.9 and 44.3 months. Other drugs achieving PFS >1 year were BRAF inhibitors [10,13] and MAPK inhibitors [6]. Larotrectinib and tazemetostat were also reported to achieve durable responses [12,23]. Eighteen patients experienced long-term disease control, defined as response (PR or CR) or stable disease (SD) for >6 months, as Table 1 demonstrates. Among them, four patients received drugs targeting BRAF mutation [6,10,13,20] and three immunotherapies [15,21], whereas the remaining patients were treated with other targeted agents. 

## 7. Discussion

In this review, we describe recent molecular studies on adult STS patients, focusing on targeted therapy implementation and its outcomes. To our knowledge, this is the first attempt to summarize the results of molecularly informed therapies in these patients to inform clinicians’ decisions. Comprehensive genomic profiling is an opportunity to address the limited therapeutic options in advanced STS patients. Our analysis reveals clinical benefits from targeted therapies in several cases, despite the fact that GIST patients were excluded. Evidence originated from retrospective series, small single-arm studies, or case reports. Thus, randomized trials are needed to evaluate this approach and compare it to standard-of-care histotype-tailored systemic therapy. A randomized French multi-center phase II/III trial is currently ongoing, assessing NGS-guided strategy versus standard treatment in metastatic sarcoma patients [41]. The French randomized SHIVA trial, which compared genomically informed therapies, targeting the hormone receptor or PAM or RME pathway over standard treatment in different tumor types, revealed improved outcomes with the latter [42]. Challenges associated with randomized trials of precision oncology have been identified, though, while conducting the MD Anderson IMPACT2 study, despite the adaptive innovative study design [43]. Improved clinical outcomes with personalized treatments across cancers have been found by a large meta-analysis of phase II single-agent studies [44].

The therapeutic relevance of molecular screening in STS is still limited. GIST constitutes the mainstay of theranostic application in sarcoma owing to the highly actionable targets *KIT* and *PDGFRa*. Apart from GIST, a driver genetic aberration that can serve as a therapeutic target has been identified in only a minority of sarcomas, including *ALK* fusion in IMT targeted with crizotinib and other TKIs [45], *PDGFb* fusion in dermatofibrosarcoma protuberans (DFSPs) targeted with imatinib [46], and *CDK4* amplification in WDLPS and DDLPS targeted with CDK4/6 inhibitors [47]. Crizotinib, in particular, led to a sustained response in one patient with *ALK*-rearranged IMT [45], a genomic aberration found in approximately 45% of IMT cases. Following this observation, the drug was also reported to be active in ALK-positive IMT [48], MET-positive alveolar soft part sarcoma [49], and MET-positive clear-cell sarcoma [50], in the biomarker-driven phase 2 EORTC 90101 CREATE trial. In addition, NTRK inhibitors are new potent tissue-agnostic targeted drugs that were recently evaluated in basket trials with tumors harboring *NTRK* gene fusions [51,52]. Larotrectinib was evaluated in three phase 1–2 trials (a phase 1 adult, a phase 1/2 pediatric, and a phase 2 adolescent/adult trial), including patients with advanced NTRK-positive solid tumors [51]. Seventy-nine percent of the patients exhibited an objective response with a favorable safety profile. Entrectinib, another NTRK inhibitor, was also evaluated in a pooled analysis of three phase 1–2 adult trials of advanced NTRK-positive solid tumors, with similar outcomes (57% objective response rate) [52]. Among these tumors, patients with infantile fibrosarcoma and other STS were included. However, the frequency of *NTRK* fusions is extremely low in adult STS, and specific histological criteria have been proposed for testing [53].

Except for the above-mentioned oncogenic drivers, the pharmacologically tractable alterations detected by NGS in STS specimens and described in this analysis are not established biomarkers that are predictive of response to a specific treatment. For instance, a proportion of leiomyosarcomas displays hallmarks of HRD and has been associated with poor clinical outcomes [17,26,54]. The PARP inhibitor olaparib was tested in combination with the chemotherapeutic drug temozolomide in heavily pretreated patients with uterine leiomyosarcoma, irrespective of genomic findings, in a small phase 2 clinical trial, showing promising results [55]. The activity of these drugs in tumors with HRD is yet to be shown. Similarly, a proportion of myxoid liposarcoma harbor PIK3CA alterations. However, the clinical efficacy of PIK3CA inhibitors in this setting will be explored in basket trials, such as NCT05307705. Another recurrent genomic finding is cell cycle gene alterations in leiomyosarcomas and many other STS [7,8,9,10,13,18,27]. Alterations in STS frequently involve more *CDKN2A/B* genes, and attempts to target them with CDK4/6 inhibitors have been made, with some anecdotal clinical benefits reported so far [9]. There is some evidence of activity of these drugs in other tumors harboring *CDKN2A* aberrations, like pancreatic cancer [56] and non-small cell lung cancer [57]. This is an example of a repurposed therapy, initially approved for hormone receptor-positive breast carcinoma. Targeting *BRAF* mutations is probably one of the most promising strategies of precision oncology in STS. Our analysis reveals a number of objective responses and prolonged PFS with BRAF inhibitors as single agents or combinational treatments, reflecting a possible oncogenic role. A recent retrospective analysis of nearly 2000 STS showed the occurrence of *BRAF* alterations in 1.2%, including mutations and fusions [58].

Apart from the challenges in identifying actionable mutations, variant allele fractions (VAFs) are often low, with an unknown impact on tumor clonality and heterogeneity. Therefore, it is challenging to make a clinical decision that relies on the presence of a genomic abnormality of unknown significance. In addition to the complexity of clinical translation, the genomic landscape of various sarcoma subtypes has been unraveled, as exemplified by specific studies on angiosarcoma [15,31], MPNST [30], and even ultra-rare histotypes, including malignant gastrointestinal neuroectodermal tumors [32] and desmoplastic small round cell tumor [33]. This will enable the screening of different biomarkers and the discovery of genetic drivers, along with the identification of novel and repurposed therapies. In addition to purely genomic alterations, other biomarkers have also emerged as potential therapeutic vulnerabilities, such as the loss of INI1 protein in epithelioid sarcoma, as a result of genetic and epigenetic events, targeted with the epigenetic modulator tazemetostat [40]. Over 90% of epithelioid sarcoma cases are characterized by loss of INI1 (also known as SMARCB1), conferring an oncogenic dependency on the enhancer of zeste homolog 2 (EZH2), the catalytic subunit of the chromatin remodeling polycomb repressive complex 2 (PRC2). Tazemetostat is a potent and highly selective EZH2 inhibitor, which showed clinical activity in epithelioid sarcoma, a chemotherapy-resistant histotype [40].

The relatively low proportion of patients that are assigned to a matched therapy across the molecular screening studies in STS is in accordance with the results of large tumor-agnostic, genomically driven, precision medicine trials, which provide access to matched targeted therapies to patients across tumor types. In the NCI-MATCH trial all over the USA, including different tumor types, only 33 of the 645 screened patients (5.1%) were assigned to treatment, of which 16 were finally enrolled in targeted therapy subprotocols [59]. The phase 2, open-label, basket trial TAPUR, led by the American Society of Clinical Oncology, includes patients with an already known targetable genetic aberration who are assigned to a matched therapy cohort [60]. The efficacy results were positive for several cohorts, including olaparib in various tumor types with *BRCA1/2*-inactivating mutations and palbociclib in *CDK4*-amplified STS, and they were negative for seven cohorts. Novel strategies of precision oncology rely on combinational treatments, instead of targeted therapies as single agents, such as the ComboMATCH trial (NCT05564377).

Molecular methods, including FISH, PCR, and more recently, NGS, have been traditionally employed in sarcoma diagnosis (Table 2). They certainly play an important role in diagnosis refinement, as illustrated by the example of *MDM2* amplification in WDLPS and DDLPS. In addition, the 2020 WHO classification of tumors of soft tissue and bone included some new entities, which can be diagnosed exclusively using molecular methods, such as sarcomas with *NTRK* fusions [61] and *BCOR* alterations [62]. Diagnostic modification based on molecular methods was found to influence therapeutic management in a small percentage of sarcoma patients [3]. NGS-based techniques were validated as a robust diagnostic tool for the detection of pathognomonic sarcoma fusion transcripts, with anchored multiplex PCR being particularly practical for routine diagnosis [63]. Genomic findings can also inform on the prognosis of different STS histotypes. For instance, synovial sarcoma with *SYT-SSX1* translocation has a worse prognosis compared to patients with *SYT-SSX2* [36]. Whole-exome sequencing of rhabdomyosarcomas led to the identification of a recurrent mutation in the *MYOD1* gene in the embryonal subtype, which is associated with a bad prognosis [64]. Clinical trials evaluating more intensive treatment protocols in these patients are warranted. It should be noted that a 67-gene expression signature was established, which predicts the clinical outcome of localized sarcomas, in order to inform clinical decisions on adjuvant treatment [65]. 

It should be noted that the results of genomic profiling largely rely on the technique and the gene panel used. Whole-genome sequencing (WGS) and whole-exome sequencing (WES) allow comprehensive genomic testing, generating a huge amount of data that are more difficult to analyze. Targeted NGS, on the other hand, uses panels of genes that are known to have a strong association with cancer or clinical actionability. WES and WGS were used in only a limited number of studies [15,16,17], whereas the majority of NGS data reported in our analysis were generated through targeted sequencing platforms, with FoundationOne^®^ being the most common, including >300 cancer-related genes. The clinical utility of the above NGS panel was compared to a limited molecular profiling panel consisting of 87 cancer-related genes in a randomized French multi-center study [18]. A subgroup analysis in sarcoma patients showed that the larger panel is associated with a higher probability of detecting an actionable alteration.

There are several limitations in the interpretation of NGS data. With regard to the occurrence of clinically actionable targets in STS, there is a variety of criteria to define a genetic alteration as actionable across the studies in STS. Additionally, different genes were interrogated on the different panels. Another challenge is the heterogeneity of STS histotypes, requiring sufficient numbers of patients in the different subgroups to draw conclusions on the genomic vulnerabilities related to specific histotypes. With regard to the efficacy of targeted therapies, a lack of drug access or availability of relevant clinical trials across countries limits the assessment of potential therapeutic targets. In some cases, there are available clinical trials for matched therapies, including only specific tumor types and not STS. In addition, it is possible that retrospective studies and case reports with only positive results are published, introducing selection bias. Unknown confounding factors may also have influenced the results, which are not derived from randomized trials, as is the case of basket trials.

Notwithstanding the limited data on targeted therapies in STS patients and the low proportion of patients who derive benefit from them to date, we feel that precision medicine through NGS is a meaningful option for patients with advanced disease. There is an urgent need for new therapies in these patients, given the poor outcomes with conventional chemotherapies. The ability to perform molecular profiling is important. As the understanding of molecular genetics evolves, new targets are being identified and efforts are being made to target some alterations known to be “undruggable”. This is the case for TP53 Y220C mutation, which is present in up to 2.9% of STS and 1.2% of bone sarcomas (rhabdomyosarcoma, leiomyosarcoma, sarcoma NOS, and osteosarcoma). The results of the phase I PYNNACLE study with PC14586, a selective inhibitor of Y220C mutant p53, capable of restoring its function, were presented last year, reporting a good tolerance and preliminary activity [66]. The phase II registration study will further assess its efficacy. In this direction, access to targeted therapies is extremely important, through either early phase and basket trials or compassionate use programs. This is currently not the case, as there are many disparities in access to NGS platforms themselves, as well as to molecularly driven therapies, across countries and regions. Efforts should be made to improve equity of access to genomic profiling and targeted therapies.

In conclusion, the advent of NGS has accelerated the discovery of effective biomarker–drug combinations in many tumors. Although comprehensive genomic profiling has focused on epithelial tumors, a growing body of evidence suggests that a number of advanced STS patients could benefit from individualized molecularly driven therapies. The results of several ongoing basket trials are expected to shed more light on the efficacy of targeted therapies in specific STS histotypes, whereas randomized trials will attempt to answer the question of whether genomic testing improves the clinical outcomes of these patients versus standard-of-care treatment. Several factors preclude the wide use of NGS in the real-world setting, including the high cost and the lack of access to molecular tumor boards and off-label therapies or clinical trials in many institutions. A deeper insight into STS biology will lead to clinical recommendations for the judicious use of these technologies in sarcoma patients based on the probability of having an actionable alteration coupled with the existence of an effective therapy. Therefore, more research is required before the transition to clinical practice.

## Figures and Tables

**Figure 1 cells-12-02632-f001:**
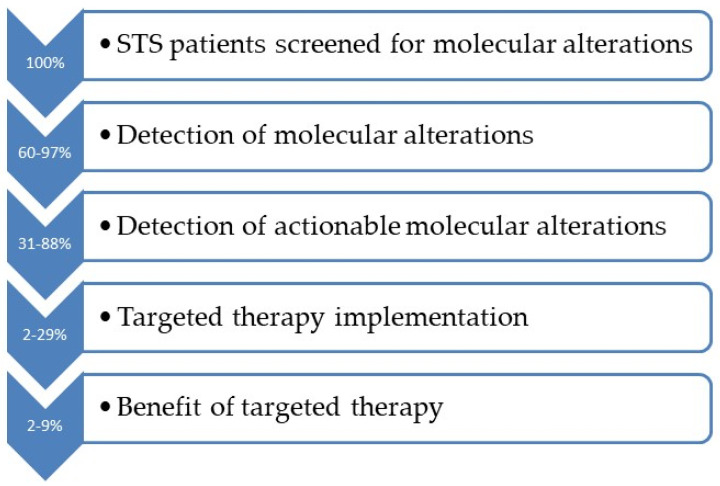
Proportion of STS patients screened for molecular alterations who eventually benefit from targeted therapies across the different studies.

**Table 1 cells-12-02632-t001:** Patients with non-GIST STS treated with molecularly recommended therapy, for which efficacy data are available.

First Author, Year	Histology	Gene Alteration(s)	Treatment	Outcome
Groisberg, 2017 [10]	Myxoid LPS	*AKT1* E17K	AKT inhibitor	SD stopped for toxicity
Arnaud-Coffin, 2020 [19]	LMS	*AKT2* ampl.	everolimus	PD, PFS 2.6 m., OS 10.9 m.
Arnaud-Coffin, 2020 [19]	UPS	*AKT2* del.	everolimus	PD, PFS 1.4 m., OS 4.1 m.
Groisberg, 2017 [10]	Pleomorphic Sa	*ALK* fus.	ceritinib	PD after 4 cycles
Subbiah et al., 2015 [22]	uIMT	*ALK* fus.	crizotinib/pazopanib	PR > 6 m.
Groisberg, 2017 [10]	Spindle cell Sa	*BRAF* fusion	sorafenib/bev/temsirol.	SD for 11 cycles
Gounder, 2022 [12]	Sa NOS	*BRAF* V600E mut.	vemurafenib + trametinib	rapid response
Groisberg, 2017 [10]	Brain GLIOSa	*BRAF* V600E mut.	vemurafenib	86% decrease, DOR 16 m.
Jin, 2021 [13]	CCS	*BRAF* V600E mut.	vemurafenib	PFS 21 m.
Lucchesi, 2018 [6]	UPS	*BRAF* V600E mut.	BRAF inhibitor	PR, PFS 7.1 m.
Massoth, 2021 [11]	HS	*BRAF* V600E mut.	dabrafenib + trametinib	rapid response, PFS 2 m.
Saijo et al., 2022 [20]	Sperm. cord Sa	*BRAF* V600E mut.	dabrafenib + trametinib	PFS 6.5 m.
Boddu, 2018 [7]	CCS	*BRAF* V600M mut.	vemurafenib	PD
Morfouace, 2023 [16]	eRMS	*BRCA1*, *BRCA2* loss	olaparib + trabectedin	PD at 2 m.
Jin, 2021 [13]	LPS	*CDK4* ampl.	palbociclib	PFS 4 m.
Gusho, 2022 [8]	LPS	*CDK4*, *MDM2*	palbociclib	PD
Gusho, 2022 [8]	LPS	*CDK4*, *MDM2*	palbociclib	SD then PD
Elvin, 2017 [9]	uLMS	*CDKN2A* mut.	palbociclib	PFS 8 m.
Gusho, 2022 [8]	Phyllodes t.	*CDKN2A*/*B*	palbociclib	PD
Gusho, 2022 [8]	Phyllodes t.	*CDKN2A*, *MTAP*	palbociclib	SD for 5 m.; PD at restart
Gusho, 2022 [8]	UPS	*CDKN2A*/*B*	palbociclib	PD
Gusho, 2022 [8]	PNST	*CDKN2A*/*B*	palbociclib	PD
Boddu, 2018 [7]	Soft parts GCT	*CDKN2A*/*B* loss	palbociclib	SD at 2 m.
Boddu, 2018 [7]	LMS	*CDKN2A*/*B* loss	palbociclib + fulvestrant	PD
Gusho, 2022 [8]	UPS	*CDKN2A*/*B*, *TP53*	palbociclib > pazopanib	PD on both drugs
Jin, 2021 [13]	FS	*COL1A1*-*PDGFB* fus.	imatinib	PFS 10 m.
Arnaud-Coffin, 2020 [19]	MPNST	*ERBB2* mut.	lapatinib	SD, PFS 1.9 m., OS 3.8 m.
Gounder, 2022 [12]	IMT	*ETV6*-*NTRK3* fus.	larotrectinib	durable CR
Recine, 2022 [23]	Spindle-cell n.	*TPM4*-*NTRK1* fus.	larotrectinib	PR, PFS 2 y.
Horak, 2021 [17]	LMS	*FGF2* fus.	pazopanib	PD, PFS 6 m.
Boddu, 2018 [7]	LMS	*FGFR1* amp.	pazopanib	PD
Boddu, 2018 [7]	UPS	*FGFR1* ampl.	pazopanib	PD
Lucchesi, 2018 [6]	RMS	*FGFR4* mut.	FGFR inhibitor	PD
Arnaud-Coffin, 2020 [19]	AS	*FLT4* ampl.	pazopanib	PR, PFS 3.1 m., OS 10.7 m.
Lucchesi, 2018 [6]	DDLPS	*FRS2* ampl.	FGFR inhibitor	SD at 5.7 m.
Lucchesi, 2018 [6]	DDLPS	*FRS2* ampl.	FGFR inhibitor	SD at 6 m.
Brahmi, 2023 [18]	MPNST	high TMB	durva + treme	CR
Gounder, 2022 [12]	UPS	high TMB	pembrolizumab	near CR
Painter et al., 2020 [15]	AS	high TMB	ICI	PFS 32.9 m.
Painter et al., 2020 [15]	AS	high TMB	ICI	PFS 44.3 m.
Boddu, 2018 [7]	UPS	*IDH1* R132C	IDH1 inhibitor	PD
Eder, 2021 [24]	lung EHE	*IDH2* mut.	olaparib	SD 11 m.
Lucchesi, 2018 [6]	LMS	*IGF1R* ampl.	mTOR inhibitor	PR
Boddu, 2018 [7]	Kaposi Sa	intermediate TMB	pembrolizumab	PR
Gounder, 2022 [12]	PEComa	intermediate TMB	nivolumab + ipilimumab	CR
Saller et al., 2018 [21]	Kaposi Sa	intermediate TMB	pembrolizumab	PFS 10.5 m.
Lucchesi, 2018 [6]	DDLPS	*KRAS* mut.	MAPK inhibitor	SD at 12.6 m.
Jin, 2021 [13]	Myofibrobl. Sa	*MAP2K1* K57N	trametinib	PFS 3 m.
Groisberg, 2017 [10]	DDLPS	*MDM2* ampl.	MDM2 inhibitor	PR x3 cycles
Groisberg, 2017 [10]	WDLPS	*MDM2* ampl.	MDM2 inhibitor	SD x8 cycles
Groisberg, 2017 [10]	WDLPS	*MDM2* ampl.	MDM2 inhibitor	CR (with resections)
Groisberg, 2017 [10]	WDLPS	*MDM2* ampl.	MDM2/MDMX inhibitor	SD for 2 cycles, toxicity
Groisberg, 2017 [10]	WDLPS	*MDM2* ampl.	MDM2 inhibitor	SD for 23 m.
Frampton, 2015 [25]	HS	*MET*ex14 alter.	crizotinib	PFS 11 m., response > 60%
Massoth, 2021 [11]	HS	*MTOR* mut.	temsirolimus/sirolimus	PFS 9 m.
Gusho, 2022 [8]	AS	*MYC*, *CUX1*	palbociclib	SD, then PD
Gusho, 2022 [8]	AS	*MYC*, *TP53*, *GNA11*	palbociclib > pazopanib	PD on both drugs
Horak, 2021 [17]	LMS	*PDGFRA* ampl.	pazopanib	PD, PFS 3.8 m.
Lopes-Brás, 2022 [14]	LPS	*PDGFRA* del.	imatinib	PR then PD, OS 2 m.
Horak, 2021 [17]	STS other	*PDGFRA*/*KIT* ampl.	pazopanib	SD, PFS 6 m.
Lopes-Brás, 2022 [14]	RMS NOS	*PIK3CA* N345I	everolimus	PR then PD, OS 4 m.
Groisberg, 2017 [10]	LMS	*PTEN* alter.	PI3K inhibitor	PD death after 3 d.
Horak, 2021 [17]	STS other	*PTPRB* mut.	pazopanib	SD, PFS 5.4 m.
Horak, 2021 [17]	LMS	*RAD18* and *BAP1* del.	olaparib + trabectedin	PFS 3 m.
Lucchesi, 2018 [6]	LMS	*RICTOR* ampl.	mTOR inhibitor	PD
Groisberg, 2017 [10]	DDLPS	*ROS1* ampl.	ceritinib	SD for 5 m.
Groisberg, 2017 [10]	LMS	*ROS1* D1538V	pazopanib + crizotinib	SD for 6 m.
Groisberg, 2017 [10]	LMS	*ROS1* D1538V	pazopanib + crizotinib	PD death prior to restaging
Boddu, 2018 [7]	AS	*ROS1* S884F	ALK/ROS/NTRK inh.	PD
Gounder, 2022 [12]	Sa NOS	*SMARCB1* del.	tazemetostat	durable PR

Ampl.: amplification, alter.: alteration, AS: angiosarcoma, bev: bevacizumab, CCS: clear cell sarcoma, CR: complete response, DDLPS: dedifferentiated liposarcoma, del.: deletion, durva: durvalumab, EHE: epithelioid hemangioendothelioma, FS: fibrosarcoma, fus.: fusion, GCT: giant cell tumor, HS: histiocytic sarcoma, IMT: inflammatory myofibroblastic tumor, LMS: leiomyosarcoma, LPS: liposarcoma, MPNST: malignant peripheral nerve sheath tumor, mut.: mutation, NOS: not otherwise specified, OS: overall survival, PD: progressive disease, PFS: progression-free survival, PR: partial response, Sa: sarcoma, SD: stable disease, TMB: tumor mutational burden, RMS: rhabdomyosarcoma, treme: tremelimumab, UPS: undifferentiated pleomorphic sarcoma, WDLPS: well-differentiated liposarcoma.

**Table 2 cells-12-02632-t002:** Most common pathognomonic genomic aberrations in STS used for diagnostic purposes.

Histotype	Gene(s)	Type of Alteration
Alveolar RMS	*PAX3*-*FOXO1*	fusion
Alveolar RMS	*PAX7*-*FOXO1*	fusion
ASPS	*TFE3*-*ASPSCR1*	fusion
Desmoid tumor	*Beta-catenin*	mutation
DFSP	*COL1A1*-*PDGFB*	fusion
DSRCT	*EWSR1*-*WT1*	fusion
EMC	*EWSR1*-*NR4A3*	fusion
ESS	*JAZF1*-*SUZ12*	fusion
ESS	*MEAF6*-*PHF1*	fusion
Ewing/PNET	*EWSR1*-*FLI1*	fusion
Ewing/PNET	*EWSR1*-*ERG*	fusion
Ewing/PNET	*EWSR1*-*FEV*	fusion
Ewing-like	*CIC*-*DUX4*	fusion
Ewing-like	*CIC*-*FOXO4*	fusion
GIST	*KIT*	mutation
GIST	*PDGFRa*	mutation
IMT	*ALK*	fusions
IMT	*ROS1*	fusions
Myxoid LPS	*FUS*-*DDIT3*	fusion
Myxoid LPS	*EWSR1*-*DDIT3*	fusion
NTRK-rearranged sarcoma	*NTRK1*, *NTRK2*, *NTRK3*	fusions
SS	*SYT*-*SSX1*	fusion
SS	*SYT*-*SSX2*	fusion
WDLPS/DDLPS	*MDM2*	amplification
WDLPS/DDLPS	*CDK4*	amplification

ASPS: alveolar soft part sarcoma, RMS: rhabdomyosarcoma, DFSP: dermatofibrosarcoma protuberans, DSRCT: desmoplastic small round cell tumor, EMC: extraskeletal myxoid chondrosarcoma, ESS: endometrial stromal sarcoma, DDLPS: dedifferentiated liposarcoma, GIST: gastrointestinal stromal tumor, IMT: inflammatory myofibroblastic tumor, LPS: liposarcoma, PNET: primitive neuroectodermal tumor, SS: synovial sarcoma, WDLPS: well-differentiated liposarcoma.

## Data Availability

Not applicable.

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
