# Peer review of "Genomic Profiling and Clinical Outcomes of Targeted Therapies in Adult Patients with Soft Tissue Sarcomas"

_cells, 2023, doi:10.3390/cells12222632_

Round 1

Reviewer 1 Report

Comments and Suggestions for Authors

The authors summarize and report on cases in which NGS has led to treatment for patients with soft-tissue sarcomas. This is very important information and deserves to be published, but we would like to add the following points to make the paper better

In Table 1, please include the NGS kit (e.g., FoundationOne) used to evaluate the genetic mutations. Also, please note whether the treatment was reimbursed by insurance , used off-label, or conducted as a clinical trial.

Author Response

In Table 1, please include the NGS kit (e.g., FoundationOne) used to evaluate the genetic mutations. Also, please note whether the treatment was reimbursed by insurance, used off-label, or conducted as a clinical trial.

Thank you very much for your point. We did not include this information in table 1, due to space limit. However, we added the following paragraph in the section #6, in which this information is provided:

FoundationOne® was the most common sequencing platform used in the studies that we analyzed [7-10,12,14,20,22,24,25]. In three studies, whole-exome sequencing was performed [15-17], whereas in two studies two different commercial NGS platforms were used [8,13]. Additionally, NGS data were generated through in-house targeted sequencing platforms in three reports [6,11,19]. Oncomine Focus Assay®, including 52 genes, was used in a case-report [23]. Finally, FoundationOne® was compared to a smaller in-house platform in the French ProfiLER 02 trial [18]. The majority of STS patients received molecularly matched therapy in a drug-matched clinical trial in [6,7,10,12,13,17,19,20,22,24]. Off-label targeted therapies were also administered in several patients [7,9,12,15-17,19,25] and a limited number of patients received targeted therapy on a compassionate use basis [17,21]. Access to matched therapy is not eluci-dated in four studies [8,11,14,18].”

Reviewer 2 Report

Comments and Suggestions for Authors

The review of Kokkali and colleagues provide an overview of ggenomic profiling and clinical outcome of targeted therapies in patients affected by soft tissue sarcoma.

The paper is interesting and has a good relevance to the field.

The manuscript would benefit from the followings:

1 In the filed of genomic profiling correlated to the clinical outcome in STS, the CINSARC study should be discussed Chibon F, Lagarde P, Salas S, Pérot G, Brouste V, Tirode F, Lucchesi C, de Reynies A, Kauffmann A, Bui B, Terrier P, Bonvalot S, Le Cesne A, Vince-Ranchère D, Blay JY, Collin F, Guillou L, Leroux A, Coindre JM, Aurias A. Validated prediction of clinical outcome in sarcomas and multiple types of cancer on the basis of a gene expression signature related to genome complexity. Nat Med. 2010 Jul;16(7):781-7. doi: 10.1038/nm.2174. Epub 2010 Jun 27. PMID: 20581836.

2 The authors should include the experience of using NGS in clinical practise for STS. In this regard a good example is the following:  Next-Generation Sequencing Approaches for the Identification of Pathognomonic Fusion Transcripts in Sarcomas: The Experience of the Italian ACC Sarcoma Working Group. Front Oncol. 2020 Apr 15;10:489. doi: 10.3389/fonc.2020.00489. Erratum in: Front Oncol. 2020 Jun 23;10:944. PMID: 32351889; PMCID: PMC7175964.

3 The authors should discuss more deeply about EZH2 inhibitor tazemetostat for epithelioid sarcoma and crizotinib for ALK.

4 Moreover the authors should include the role of NTRK1-2-3 inhibitors entrectenib and larotrectnib in emerging STS histotypes. In this regard the following manuscript should be included: Case Report: Adult NTRK-Rearranged Spindle Cell Neoplasm: Early Tumor Shrinkage in a Case With Bone and Visceral Metastases Treated With Targeted Therapy. Front Oncol. 2022 Jan 7;11:740676. doi: 10.3389/fonc.2021.740676. PMID: 35070960; PMCID: PMC8776642.

5 Future directions should be included

Author Response

Thank you very much for your informative points.

The manuscript would benefit from the followings:

1 In the filed of genomic profiling correlated to the clinical outcome in STS, the CINSARC study should be discussed Chibon F, Lagarde P, Salas S, Pérot G, Brouste V, Tirode F, Lucchesi C, de Reynies A, Kauffmann A, Bui B, Terrier P, Bonvalot S, Le Cesne A, Vince-Ranchère D, Blay JY, Collin F, Guillou L, Leroux A, Coindre JM, Aurias A. Validated prediction of clinical outcome in sarcomas and multiple types of cancer on the basis of a gene expression signature related to genome complexity. Nat Med. 2010 Jul;16(7):781-7. doi: 10.1038/nm.2174. Epub 2010 Jun 27. PMID: 20581836.

We commented on this study in the discussion session, in the paragraph concerning diagnostic and prognostic application of genomics in sarcoma:

It should be noted that a 67-gene expression signature was established, which predicts clinical outcome of localized sarcomas, in order to inform clinical decisions on adjuvant treatment [65].”

2 The authors should include the experience of using NGS in clinical practise for STS. In this regard a good example is the following:  Next-Generation Sequencing Approaches for the Identification of Pathognomonic Fusion Transcripts in Sarcomas: The Experience of the Italian ACC Sarcoma Working Group. Front Oncol. 2020 Apr 15;10:489. doi: 10.3389/fonc.2020.00489. Erratum in: Front Oncol. 2020 Jun 23;10:944. PMID: 32351889; PMCID: PMC7175964.

Thank you very much for your fruitful comment. Indeed NGS is more useful for diagnostic purposes in clinical practice for sarcoma patients. This point is being developed in the discussion section in the following paragraph:

Molecular methods, including FISH, PCR and more recently NGS, have been tradi-tionally employed in sarcoma diagnosis (table 2). They certainly play an important role in diagnosis refinement, as illustrated by the example of MDM2 amplification in WDLPS and DDLPS. In addition, the 2020 WHO classification of Tumours of Soft Tissue and Bone included some new entities, which can be diagnosed exclusively using molecular methods, such as sarcomas with NTRK fusions [6157] and BCOR alterations [6258]. Diagnostic modification based on molecular methods was found to influence the therapeutic management in a small percentage of sarcoma patients [3]

To emphasize on the clinical utility of NGS methods in sarcoma diagnosis, we added the following part in the above-mentioned paragraph, commenting on the paper of the Italian Sarcoma Group (Racanelli et al., 2020):

NGS-based techniques were validated as a robust diagnostic tool for the detection of pathognomonic sarcoma fusion transcripts, with anchored multiplex PCR being particularly practical for routine diagnosis [63].”

3 The authors should discuss more deeply about EZH2 inhibitor tazemetostat for epithelioid sarcoma and crizotinib for ALK.

We added the following parts in the discussion:

-in the 4th paragraph (about genomic alterations in specific histotypes):  

Over 90% of epithelioid sarcoma cases are characterized by loss of INI1 (also known as SMARCB1), conferring an oncogenic dependency on enhancer of zeste homolog 2 (EZH2), the catalytic subunit of the chromatin remodelling polycomb repressive complex 2 (PRC2). Tazemetostat is a potent and highly selective EZH2 inhibitor, which showed clinical activity in epithelioid sarcoma, a chemotherapy-resistant histotype [40].

-in the 2th paragraph (about established oncogenic drivers in STS):

Crizotinib, in particular, led to a sustained response in one patient with ALK-rearranged IMT [45], a genomic aberration found in approximately 45% of IMT cases. Following this observation, the drug was also reported to be active in ALK-positive IMT [48], MET-positive alveolar soft part sarcoma [49] and MET-positive clear-cell sarcoma [50], in the biomarker-driven phase 2 EORTC 90101 CREATE trial.”

4 Moreover the authors should include the role of NTRK1-2-3 inhibitors entrectenib and larotrectnib in emerging STS histotypes. In this regard the following manuscript should be included: Case Report: Adult NTRK-Rearranged Spindle Cell Neoplasm: Early Tumor Shrinkage in a Case With Bone and Visceral Metastases Treated With Targeted Therapy. Front Oncol. 2022 Jan 7;11:740676. doi: 10.3389/fonc.2021.740676. PMID: 35070960; PMCID: PMC8776642.

The recommended case-report was added in the table 1.

We added the following part in the 2th paragraph of the discussion (about established oncogenic drivers in STS):

 “Larotrectinib was evaluated in three phase 1-2 trials (a phase 1 adult, a phase 1/2 pediatric and a phase 2 adolescent/adult trial) including patients with advanced NTRK-positive solid tumors [51]. Seventy-nine percent of the patients exhibited an objective response, with a favorable safety profile. Entrectinib, another NTRK inhibitor, was also evaluated in a pooled analysis of three phase 1-2 adult trials of advanced NTRK-positive solid tumors, with similar outcomes (57% objective response rate) [52].”

5 Future directions should be included

We added a paragraph on future directions, in the discussion section just before the conclusion:

Notwithstanding the limited data on targeted therapies in STS patients and the low proportion of patients who derive benefit from them to date, we feel that precision medicine through NGS is a meaningful option for patients with advanced disease. There is an urgent need for new therapies in these patients, given the poor outcomes with conventional chemotherapies. The ability to perform molecular profiling is important, as the understanding of molecular genetics evolves, new targets are being identified and efforts are being made to target some alterations known to be “undruggable”. This is the case for TP53 Y220C mutation, which is present in up to 2.9% of STS and 1.2% of bone sarcomas (rhabdomyosarcoma, leiomyosarcoma, sarcoma NOS and osteosarcoma). The results of the phase I PYNNACLE study with PC14586, a selective inhibitor of Y220C mutant p53, capable of restoring its function, were presented last year, reporting a good tolerance and preliminary activity [66]. The phase II registration study will further assess its efficacy. In this direction, access to targeted therapies is extremely important, through either early phase and basket trials or compassionate use programs. This is currently not the case, as there are many disparities in access to NGS platforms themselves, as well as to molecularly driven therapies, across countries and regions. Efforts should be made to improve equity of access to genomic profiling and targeted therapies.”